# Multimodal Representation Learning by Alternating Unimodal Adaptation

## Abstract

Multimodal learning, which integrates data from diverse sensory modes, plays a pivotal role in artificial intelligence. However, existing multimodal learning methods often struggle with challenges where some modalities appear more dominant than others during multimodal learning, resulting in suboptimal performance. To address this challenge, we propose MLA (Multimodal Learning with Alternating Unimodal Adaptation). MLA reframes the conventional joint multimodal learning process by transforming it into an alternating unimodal learning process, thereby minimizing interference between modalities. Simultaneously, it captures cross-modal interactions through a shared head, which undergoes continuous optimization across different modalities. This optimization process is controlled by a gradient modification mechanism to prevent the shared head from losing previously acquired information. During the inference phase, MLA utilizes a test-time uncertainty-based model fusion mechanism to integrate multimodal information. Extensive experiments are conducted on five diverse datasets, encompassing scenarios with complete modalities and scenarios with missing modalities. These experiments demonstrate the superiority of MLA over competing prior approaches.

## 1 Introduction

Multimodal learning, which draws inspiration from the multi-sensory perception mechanisms in humans, has gained significant prominence in the field of artificial intelligence (Tan & Bansal, 2019; Yu et al., 2023; Sung et al., 2023). However, recent multimodal learning methods often struggle to fully integrate rich multimodal knowledge across different modalities, and we argue that a key factor is *modality laziness*. In multimodal representation learning, some modalities are more dominant than others (Peng et al., 2022; Du et al., 2023a), so the model will optimize for these dominant modalities and tend to ignore others, resulting in suboptimal performance (Ismail et al., 2020a; Sun et al., 2021; Wang et al., 2020a). This is because collected multimodal data are often not well entangled with each other, or their data size varies. In a more extreme scenario, critical modality data may be missing depending on the conditions during the data collection phase (Lian et al., 2022). This is particularly one of the main challenges in multimodal learning on uncurated real-world data.

A few recent works have been introduced to balance the influence of dominating versus subordinate modalities in the optimization process (Peng et al., 2022; Zhang et al., 2023). However, these methods necessitate joint optimization of different modes to update the multiple modality-specific encoders simultaneously, which degenerates the adaptation for subordinate modalities to some extent, thereby limiting overall multi-modal performance (Wang et al., 2020a). In contrast, we aim to tackle this problem in a conceptually different way by decomposing the conventional multimodal joint optimization scenario into an alternating unimodal learning scenario, leading to an approach named **M**ultimodal **L**earning with **A**lternating Unimodal Adaptation (**MLA**). The key idea of MLA is to alternately optimize the encoder of each modality, while simultaneously integrating cross-modal information.

Concretely, as shown in Figure 1, the predictive function of each modality in our approach includes a modality-specific encoder and a shared head across all modalities. In the alternating unimodal learning paradigm, the predictive functions for each modality are optimized alternately to eliminate interference across modalities. Simultaneously, the shared head is optimized continuously across modalities, essentially capturing cross-modal information. However, in this optimization process, the

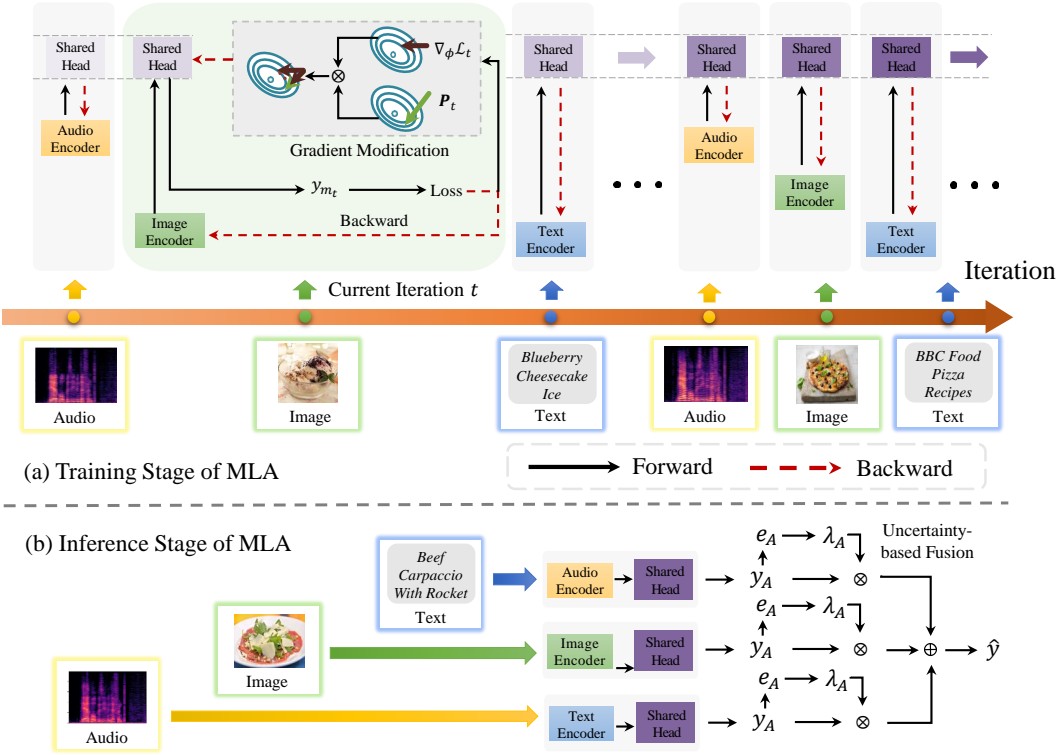

Figure 1: The framework of MLA: (a) Training stage, where we employ an alternating unimodal adaptation process, processing only one modality at each iteration. The shared head captures cross-modal information, and we also introduce gradient modification to prevent forgetting learned modality information from previous iterations; (b) Testing stage, where we introduce an uncertainty-based test-time multimodal fusion approach to combine multimodal information.

head is susceptible to losing previously learned information from other modalities when it encounters a new modality, which is referred to as modality forgetting. To address this issue, we introduce a gradient modification mechanism for the shared head to encourage the orthogonalization of gradient directions between modalities. After learning the modality-specific encoders and the shared head, we further propose a test-time dynamic modality fusion mechanism to integrate multimodal information. Since there are information gaps among different modalities contributing to the prediction, we evaluate the significance of each modality and use this evaluation to assign weights to the predictions generated by each modality. Our method for gauging the importance of each modality relies on measuring the level of uncertainty observed in the predictions associated with that modality. This mechanism is motivated by the hypothesis that when one modality exhibits higher uncertainty in its predictions, it is more prone to producing incorrect predictions.

Our primary contribution of this paper is MLA, which introduces an innovative alternating unimodal optimization approach in multimodal learning. This approach not only enables relatively independent optimization within each modality but also preserves cross-modal interactions. MLA is also compatible with scenarios involving both complete and missing modalities in the learning process. In both scenarios, our empirical results demonstrate the promise of MLA in addressing modality laziness and enhancing multimodal learning performance compared to the best prior methods. Furthermore, we show that MLA can enlarge the modality gap, offering further insights into the performance improvements it achieves.

## 2    MULTIMODAL LEARNING WITH ALTERNATING UNIMODAL ADAPTATION

This section presents our proposed method to address the modality laziness issue, named *Multimodal Learning with Alternative Unimodal Adaptation* (**MLA**). Motivated by the challenge of information

imbalance in different modalities, MLA aims to reframe conventional joint training scheme for multimodal data into the context of a alternating unimodal learning framework, as illustrated in Figure 1. Specifically, during the training phase (Figure 1(a)), our approach alternately learns the encoders for each modality while simultaneously maintaining cross-modal information using a shared head. We introduce a gradient modification mechanism to prevent the shared head from forgetting previously learned modality information. During inference, our approach dynamically fuses multimodal information by evaluating the uncertainty of the prediction for each modality (Figure 1(b)). Next, we will introduce the three key stages: *alternating unimodal learning*, *learning cross-modal information without modality forgetting*, and *test-time dynamic modality fusion*.

## 2.1 ALTERNATING UNIMODAL LEARNING

In multimodal learning scenarios, the phenomenon known as modality laziness arises due to the inadequate optimization of less informative modalities when these are learned in conjunction with others, subsequently leading to suboptimal fusion performance. To tackle this problem, we propose an alternating unimodal learning paradigm. Here, each modality undergoes an independent optimization process, eliminating interference across modalities, thus allowing each modality to reach its full potential without being overshadowed by more informative ones.

Specifically, in a multimodal learning scenario, consider the presence of a total of $M$ modalities, each corresponding to a dataset denoted as $\mathcal{D}_m$, with $\mathcal{D}_m = (X_m, Y_m) = \{(x_{m,k}, y_{m,k})\}_{k=1}^{N_m}$, where $N_m$ represents the number of examples in modality $m$. Additionally, each modality $m$ is associated with a predictive function formulated as $f_m = g \circ h_m$, wherein the function $h_m$ serves as the modality-specific encoder and the function $g$ denotes the shared head across all modalities. Given $T$ total training iterations, for every iteration $t$ where $t > M$, the model receives data exclusively from one modality. The input modality, denoted as $m_t$, for iteration $t$ is calculated as follows:

$$m_t = t \bmod M. \tag{1}$$

After determining the modality type $m_t$ for each iteration $t$, the next step is to train the modality-specific function $f_{m_t}$. This is achieved by minimizing the predictive risk for every example within the corresponding dataset $\mathcal{D}_{m_t}$ when it is processed through the corresponding predictive function $f_{m_t}$. The loss during iteration $t$ is formally defined as:

$$\mathcal{L}_t = \mathbb{E}_{(x,y) \sim \mathcal{D}_{m_t}}[\ell(f_{m_t}(x; \theta_{m_t}, \phi), y)] = \mathbb{E}_{(x,y) \sim \mathcal{D}_{m_t}}[\ell(g(h_{m_t}(x; \theta_{m_t}); \phi), y)]. \tag{2}$$

where $\theta_{m_t}$ and $\phi$ are learnable parameters of encoder $h_{m_t}$ and head $g$, respectively.

According to equation 2, in every iteration $t$, data is exclusively fed from a single modality, and the optimization is solely on the corresponding predictive function. In this way, MLA prevents any information imbalance between modalities during the optimization process, thereby avoiding modality laziness. It's also worth noting that MLA does not require paired multimodal data during the training phase, making it a natural fit for scenarios with extreme modality laziness, such as learning with missing modalities.

## 2.2 LEARNING CROSS-MODAL INFORMATION WITHOUT MODALITY FORGETTING

In multimodal learning, besides isolating the optimization process for each modality to prevent modality laziness, capturing cross-modal interaction information stands out as another crucial aspect. As per Equation 2, the shared head $g$ undergoes optimization at every iteration, enabling the capture of cross-modal interaction information throughout the process. Nonetheless, this optimization phase for head $g$ poses a significant challenge: the head $g$ is prone to forgetting the information of previously trained modalities when learning the current one, a phenomenon termed as *modality forgetting*. This issue can significantly undermine the effectiveness of cross-modal information learning.

To resolve this issue, inspired by the orthogonal weight modification (Zeng et al., 2019), we introduce a gradient modification matrix $\mathbf{P}_t$ to adjust the weights $\phi$ of head $g$ before each optimization iteration $t$. This gradient modification ensures that the parameter update direction is orthogonal to the plane spanned by the encoded feature from the previous modality. The gradient modification matrix can be computed using Recursive Least Squares algorithms. Specifically, at each iteration $t$, the

---

**Algorithm 1:** Multimodal Learning with Alternating Unimodal Adaptation (**MLA**)

---

**Input:** Multimodal datasets $\mathcal{D}_1, \mathcal{D}_2, \ldots, \mathcal{D}_M$, # of training iterations $T$

1 **for** $t = 1$ **to** $T$ **do**
2     Calculate the modality assignment $m_t$ at iteration $t$ as Equation 1;
3     Compute the loss $\mathcal{L}_t$ following Equation 2 with $m_t$;
4     Update the modality-specific parameters $\theta_{m_t}$ with gradient descent;
5     Compute the gradient modification matrix $\mathbf{P}_t$ as described in Equation 3;
6     Update the shared parameters $\phi_t$ using Equation 4;

7 **Inference stage:**
8 **for** *Test example $(x_r, y_r)$ involving $M$ modalities* **do**
9     Calculate the prediction uncertainty $e_{m,r}$ for each modality $m$ following Equation 6;
10     Determine the modality importance coefficients $\lambda_{m,r}$ using Equation 7;
11     Combine the predictions from all modalities as Equation 5 and get the predicted value $\hat{y}_r$;

---

corresponding modification matrix $\mathbf{P}_t$ is calculated as follows:

$$\mathbf{P}_t = \mathbf{P}_{t-1} - \mathbf{q}_t \left[ \overline{h_{m_t}(x)} \right]^T \mathbf{P}_{t-1},$$

$$\text{where } \mathbf{q}_t = \frac{\mathbf{P}_t \overline{h_{m_t}(x)}}{\alpha + \left[ \overline{h_{m_t}(x)} \right]^T \mathbf{P}_{t-1} \overline{h_{m_t}(x)}}, \quad \overline{h_{m_t}(x)} = \frac{1}{N_{m_t}} \sum_{k=1}^{N_{m_t}} h_{m_t}(x_{m_t,k}), \tag{3}$$

where $\alpha$ is a predefined hyperparameter used to prevent denominator from being zero. The $\mathbf{P}_t \in \mathbb{R}^{s \times s}$, where $s$ represents the dimension of the output from the encoder. Based on the definition of $\mathbf{P}_t$, we update the parameters $\phi$ of the shared head $g$ during iteration $t$ as follows:

$$\phi_t = \phi_{t-1} - \gamma \begin{cases} \nabla_\phi \mathcal{L}_t & \text{if } t = 0, \\ \mathbf{P}_t \nabla \phi \mathcal{L}_t & \text{if } t > 0, \end{cases} \tag{4}$$

where $\mathcal{L}_t$ is defined in equation 2. By introducing the gradient orthogonalization process to calibrate the weight update direction for the shared head $g$ across consecutive modalities, we ensure that applying the gradients to one modality minimally interferes with its prior modality, which facilitates more effective capture of cross-modal information.

## 2.3 TEST-TIME DYNAMIC MODALITY FUSION

After learning the modality-specific encoders and the shared head during the training process, we focus on how to effectively integrate multimodal information to make prediction during inference time. To achieve this multimodal fusion, we employ a weighted combination of predictions from each modality. Specifically, for a given test example $(x_r, y_r)$, which involves $M$ modalities, we calculate its prediction as follows:

$$\hat{y}_r = \sum_{m=1}^{M} \lambda_{m,r} f_m(x_{m,r}; \theta_m^*, \phi^*), \tag{5}$$

where $\lambda_{m,r}$ signifies the importance of modality $m$ in predicting the labels for the test example $r$. The parameters $\theta_m^*$ and $\phi^*$ are optimized values associated with the encoder $h_m$ and the shared head $g$, respectively.

To determine the value of modality importance coefficient $\lambda_m$, MLA performs under the hypothesis that when one modality exhibits higher uncertainty in its predictions, it becomes more prone to making incorrect predictions. Consequently, we leverage prediction uncertainty as a proxy to gauge the importance of each modality. Specifically, we begin by assessing the uncertainty $e_{m,r}$ using entropy of each individual modality's output as follows:

$$e_{m,r} = -p_{m,r}^T \log p_{m,r}, \text{ where } p_{m,r} = \text{Softmax}(f_m(x_{m,r}; \theta_m^*, \phi^*)). \tag{6}$$

Here, $\text{Softmax}(\cdot)$ converts the output logits to the probability $p_{m,r}$. A higher entropy $e_{m,r}$ indicates lower confidence in the prediction, leading to a smaller importance weight during the fusion process.

Based on this, we calculate the importance weight for a modality $m$ as:

$$\lambda_{m,r} = \frac{\exp\left(\max\limits_{m=1,...,M} e_{m,r} - e_m\right)}{\sum_{v=1}^{M} \exp\left(\max\limits_{m=1,...,M} e_{m,r} - e_{v,r}\right)}. \tag{7}$$

By introducing the test-time dynamic fusion mechanism that explicitly considers the predictive uncertainty associated with each modality, MLA is better equipped to handle scenarios with imbalances in modality-specific information, enhancing the effectiveness of multimodal fusion. The whole training and inference pipeline is provided in Algorithm 1.

## 3 EXPERIMENTS

In this section, we evaluate the performance of MLA, aiming to answer the following questions: **Q1:** Compared to prior approaches, can MLA overcome modality laziness and improve multimodal learning performance? **Q2:** How does MLA perform when faced the challenge of missing modalities? **Q3:** Can the proposed modules (e.g., test-time dynamic fusion) effectively improve performance? **Q4:** How does MLA change the modality gap in multimodal learning?

### 3.1 LEARNING WITH COMPLETE MODALITIES

#### 3.1.1 EXPERIMENTAL SETUPS

**Datasets.** We utilize a set of five datasets with different tasks to assess the performance of learning with complete modalities (see Appendix A.1 for detailed data statistics and descriptions). **(1)&(2) CREMA-D** (Cao et al., 2014) **and Kinetic-Sound (KS)** (Arandjelovic & Zisserman, 2017) belong to the category of *audio-video* datasets. CREMA-D provides audio and video recordings depicting various emotions within the Haitian Creole culture, while KS combines video and audio data for object and action recognition. **(3)&(4) Food-101** (Wang et al., 2015b) **and MVSA** (Niu et al., 2016) are both *image-text* datasets. Food-101 comprises over 100,000 food images accompanied by corresponding texts, with a focus on food classification. MVSA concentrates on sentiment classification in multimedia posts through the utilization of both text and image data. **(5) IEMOCAP** (Busso et al., 2008) is a *audio-image-text* dataset that captures emotions across audio, vision, and text data during natural conversations.

**Baselines.** We conducted a comprehensive comparison of MLA with (1) *conventional multimodal fusion methods*: including summation (Sum), concatenation (Concat), late fusion (Gunes & Piccardi, 2005); (2) *modulation-based fusion methods*: including FiLM (Perez et al., 2018), and BiLinear Gated (BiGated) (Kiela et al., 2018); (3) *methods for addressing modality-laziness*: including OGM-GE (Peng et al., 2022) and QMF (Zhang et al., 2023). For more detailed baseline descriptions, please refer to Appendix A.2.

**Backbone and Hyperparameter Settings.** For the audio-video task (CREMA-D and KS datasets), we employ a ResNet-18-based (Peng et al., 2022) network as the encoder. In the image-text task (Food-101 and MVSA), we utilize M3AE (Geng et al., 2022), a large pre-trained multimodal masked auto-encoder, as the encoder. For the audio-image-text task (IEMOCAP), we integrate M3AE with another large pre-trained model named CAVMAE (Gong et al., 2023) as the encoder. In all experiments, a fully connected layer served as the shared head across different modalities. To ensure a fair comparison, all baselines used the same backbone architectures. We determined all other hyperparameters through cross-validation. Further details regarding our experimental setups can be found in Appendix A.3.

#### 3.1.2 RESULT

In Table 1, we report the results of only using a single modality and the results achieved by combining all modalities. The observations from Table 1 reveal several key insights: *Firstly*, most conventional fusion and modulation-based methodologies, with the exception of late fusion, faced challenge of modality laziness. This was evidenced by a noticeable performance disparity on numerous datasets between the superior and inferior modality performances. *Second*, the late fusion approach addresses

Table 1: Results on audio-video (A-V), image-text (I-T), and audio-image-text (A-I-T) datasets. Both the results of only using a single modality and the results of combining all modalities ("Multi") are listed. We report the average test accuracy (%) of three random seeds. Full results with standard deviation are reported in Appendix A.4. The best results and second best results are **bold** and underlined, respectively.

| Type | Data | | Sum | Concat | Late Fusion | FiLM | BiGated | OGM-GE | QMF | **MLA (Ours)** |
|------|------|------|-----|--------|-------------|------|---------|--------|-----|----------------|
| A-V | CREMA-D | Audio | 54.14 | 55.65 | 52.17 | 53.89 | 51.49 | 53.76 | **59.41** | 59.27 |
| | | Video | 18.45 | 18.68 | 55.48 | 18.67 | 17.34 | 28.09 | 39.11 | **64.91** |
| | | Multi | 60.32 | 61.56 | 66.32 | 60.07 | 59.21 | 68.14 | 63.71 | **79.70** |
| | KS | Audio | 48.77 | 49.18 | 47.87 | 48.67 | 49.96 | 48.87 | 51.57 | **54.67** |
| | | Video | 24.53 | 24.67 | 46.76 | 23.15 | 23.77 | 29.73 | 32.19 | **51.03** |
| | | Multi | 64.72 | 64.84 | 65.53 | 63.33 | 63.72 | 65.74 | 65.78 | **71.35** |
| I-T | Food-101 | Image | 4.57 | 3.51 | 58.46 | 4.68 | 14.20 | 22.35 | 45.74 | **69.60** |
| | | Text | 85.63 | 86.02 | 85.19 | 85.84 | 85.79 | 85.17 | 84.13 | **86.47** |
| | | Multi | 86.19 | 86.32 | 90.21 | 87.21 | 88.87 | 87.54 | 92.87 | **93.33** |
| | MVSA | Text | 73.33 | 75.22 | 72.15 | 74.85 | 73.13 | 74.76 | 74.87 | **75.72** |
| | | Image | 28.46 | 27.32 | 45.24 | 27.12 | 28.15 | 31.98 | 32.99 | **54.99** |
| | | Multi | 76.19 | 76.25 | 76.88 | 75.34 | 75.94 | 76.37 | 77.96 | **79.94** |
| A-I-T | IEMOCAP | Audio | 39.79 | 41.93 | 43.12 | 41.64 | 42.23 | 41.38 | 42.98 | **46.29** |
| | | Image | 29.44 | 30.00 | 32.38 | 29.85 | 27.45 | 30.24 | 31.22 | **37.63** |
| | | Text | 65.16 | 67.84 | 68.79 | 66.37 | 65.16 | 70.79 | 75.03 | **73.22** |
| | | Multi | 74.18 | 75.91 | 74.96 | 74.32 | 73.34 | 76.17 | 76.17 | **78.92** |

modality laziness by training each modality's encoder exclusively on the corresponding unimodal data. However, while late fusion mitigates modality laziness to some extent, it falls short in delivering satisfactory performance when integrating information from all modalities. This limitation primarily stems from its inability to effectively capture cross-modal information. *Third*, both OGM-GE and QMF contribute to reducing modality laziness and enhancing multimodal performance to some degree. However, they do not completely bridge the gap between modalities. *Finally*, MLA instead consistently outperforms all other methods across all scenarios. This demonstrates MLA's ability to effectively address modality laziness and enhance multimodal learning performance by fully leveraging the information from each modality and capturing cross-modal knowledge.

## 3.2 LEARNING WITH MISSING MODALITIES

### 3.2.1 EXPERIMENTAL SETUP

**Evaluation Strategy.** Besides learning with complete modalities, we further evaluate the performance on datasets with missing modalities, which could be regarded as an extreme case of modality laziness. Here, follow (Lian et al., 2022), we apply percentage-based masks to both the training and testing data within the IEMOCAP dataset. Specifically, we randomly mask each modality of each sample with a probability $\eta$. Following Yuan et al. (2021); Zhang et al. (2022b), we select missing rates from the list $[0.1, 0.2, ...0.7]$, maintaining the same missing rate across the training, validation, and testing phases.

**Baselines and Hyperparameter Settings.** We compare MLA with the following baselines: (1) the strongest and most compatible baselines in the experiments of learning with complete modalities, including late fusion (Gunes & Piccardi, 2005) and QMF (Zhang et al., 2023); (2) methods specifically designed for learning with missing modalities, including CCA (Hotelling, 1992), DCCA (Andrew et al., 2013), DCCAE (Wang et al., 2015a), AE (Bengio et al., 2006), CRA (Tran et al., 2017), MMIN (Zhao et al., 2021), IF-MMIN (Zuo et al., 2022), CPM-Net (Zhang et al., 2022a), and TATE (Zeng et al., 2022). All baselines use the same backbone models as used in the experiments of learning with complete modalities. We tune the hyperparameters via cross-validation. Detailed baseline descriptions and experimental setups can be found in Appendix A.3.

### 3.2.2 RESULTS

We report the results in Table 2, showing the performance under different modality missing rates. We observe that (1) all approaches show performance degradation with the increase of modality missing

Table 2: We report the test accuracy percentages (%) on the IEMOCAP dataset using three different seeds, while applying varying modality missing rates to audio, image, and text data. The best results are highlighted in **bold**, while the second-best results are underlined.

| Method | Modality Missing Rate (%) | | | | | | |
|---|---|---|---|---|---|---|---|
| | 10 | 20 | 30 | 40 | 50 | 60 | 70 |
| Late Fusion | 72.95 | 69.06 | 64.89 | 61.09 | 56.48 | 52.41 | 45.07 |
| QMF | 73.49 | 71.33 | 65.89 | 62.27 | 57.94 | 55.60 | 50.25 |
| CCA | 65.19 | 62.60 | 59.35 | 55.25 | 51.38 | 45.73 | 30.61 |
| DCCA | 57.25 | 51.74 | 42.53 | 36.54 | 34.82 | 33.65 | 41.09 |
| DCCAE | 61.66 | 57.67 | 54.95 | 51.08 | 45.71 | 39.07 | 41.42 |
| AE | 71.36 | 67.40 | 62.02 | 57.24 | 50.56 | 43.04 | 39.86 |
| CRA | 71.28 | 67.34 | 62.24 | 57.04 | 49.86 | 43.22 | 38.56 |
| MMIN | 71.84 | 69.36 | 66.34 | 63.30 | 60.54 | 57.52 | 55.44 |
| IF-MMIN | 71.32 | 68.29 | 64.17 | 60.13 | 57.45 | 53.26 | 52.04 |
| CPM-Net | 55.29 | 53.65 | 52.52 | 51.01 | 49.09 | 47.38 | 44.76 |
| TATE | 67.84 | 63.22 | 62.19 | 60.36 | 58.74 | 57.99 | 54.35 |
| **MLA (Ours)** | **75.07** | **72.33** | **68.47** | **67.00** | **63.48** | **59.17** | **55.89** |

rate. This is what we expected because utilizing all modality data tends to enhance performance when compared to using only partial modality data, as also corroborated by the results in Table 1, where employing multi-modality information outperforms using only single-modality information; (2) MLA consistently outperforms the other baselines across all missing rates, including general methods (Late Fusion and QMF) and methods that are specifically designed for tackling missing modality (e.g., MMIN). These results highlight the effectiveness of MLA in addressing the challenge of modality laziness, even under the extreme case of learning with missing modality. This further underscores the effectiveness of our proposed alternating unimodal learning and test-time dynamic modality fusion mechanisms.

## 3.3 ANALYSIS OF MLA

In this section, we aim to gain a deeper insight into the performance improvements achieved by MLA. To do so, we conduct four analyses. Firstly, we conduct an ablation study, which focuses on head gradient modification and test-time modality fusion. Secondly, we explore the isolation of modalities within our alternating unimodal learning framework. Third, we analyze the modality gap. Lastly, we conduct experiments to analyze the robustness of ours over pre-trained models.

### 3.3.1 ABLATION STUDY

We first conduct ablation studies to demonstrate the effectiveness of using head gradient modification in alternating unimodal learning and test-time dynamic fusion. We report the results in Table 3 and observe that: (1) Modifying head weights under the alternate unimodal learning framework enhances the performance of both

Table 3: Results of ablation studies on five datasets. We report the average test accuracy (%) of three random seeds. Full results with standard deviation are reported in Appendix A.4. HGM: head gradient modification; DF: dynamic fusion.

| Data | | HGM | DF | HGM | DF | HGM | DF | HGM | DF |
|---|---|---|---|---|---|---|---|---|---|
| | | ✗ | ✗ | ✗ | ✓ | ✓ | ✗ | ✓ | ✓ |
| CREMA-D | Audio | 52.17 | | 52.17 | | **59.27** | | **59.27** | |
| | Video | 55.48 | | 55.48 | | **64.91** | | **64.91** | |
| | Multi | 66.32 | | 72.79 | | 74.51 | | **79.70** | |
| KS | Audio | 47.87 | | 47.87 | | **54.66** | | **54.66** | |
| | Video | 46.76 | | 46.76 | | **51.03** | | **51.03** | |
| | Multi | 65.53 | | 66.34 | | 70.72 | | **71.35** | |
| Food-101 | Text | 85.19 | | 85.19 | | **86.47** | | **86.47** | |
| | Image | 58.46 | | 58.46 | | **69.60** | | **69.60** | |
| | Multi | 90.21 | | 91.37 | | 91.72 | | **93.33** | |
| MVSA | Text | 72.15 | | 72.15 | | **75.72** | | **75.72** | |
| | Image | 45.24 | | 45.24 | | **54.99** | | **54.99** | |
| | Multi | 76.88 | | 77.53 | | 79.59 | | **79.94** | |
| IEMOCAP | Audio | 43.12 | | 43.12 | | **46.29** | | **46.29** | |
| | Text | 68.79 | | 68.79 | | **73.22** | | **73.22** | |
| | Image | 32.38 | | 32.38 | | **37.63** | | **37.63** | |
| | Multi | 74.96 | | 75.42 | | 77.58 | | **78.92** | |

using the information from a single modality only and multiple modalities ("Multi"). This improvement is expected, as mitigating the effect of modality forgetting enables better integration of cross-modal information, benefiting both unimodal and multimodal learning processes; (2) Using a test-time dynamic fusion mechanism can significantly advance the multimodal learning process. This result is anticipated, as test-time dynamic fusion accounts for modal predictive uncertainty, which usually correlates with model performance. By using predictive uncertainty to measure the

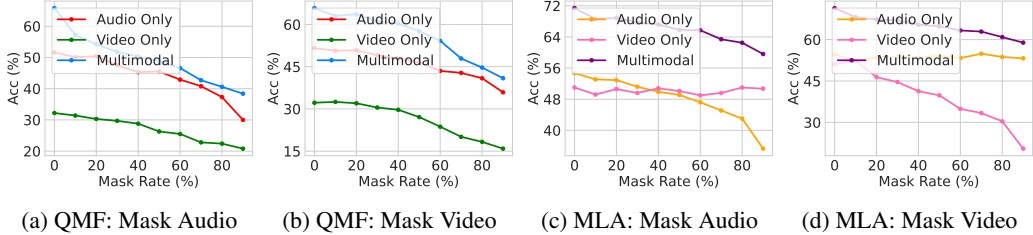

| (a) QMF: Mask Audio | (b) QMF: Mask Video | (c) MLA: Mask Audio | (d) MLA: Mask Video |

Figure 2: Visualization of test accuracy (%) on the KS dataset, varying with the ratio of missing modality in either audio or video training data.

modality importance on test examples, the model can more accurately fuse information from multiple modalities; (3) By integrating both strategies, MLA exhibits the best performance, demonstrating its effectiveness in multimodal learning.

### 3.3.2 ANALYSIS OF MODALITY ISOLATION IN ALTERNATING UNIMODAL LEARNING

We investigate how the alternating unimodal learning paradigm isolates the unimodal learning and prevents modality laziness. In this context, we selectively masked either the audio or video modality during training, employing various modality masking ratios. The results are presented in Figure 2, where the results on QMF are also illustrated for comparison. We observe that the performance of unimodal learning in MLA remains unaffected by the absence of other modalities. In contrast, in QMF, the absence of one modality negatively impacts the performance of another modality. These findings further strengthen our argument that adopting an alternating optimization approach to learn the predictive functions of different modalities can effectively address the modality laziness issue.

### 3.3.3 ANALYSIS OF MODALITY GAP

As demonstrated in Liang et al. (2022), there exists a modality gap in multimodal learning, wherein different modality information is situated in two entirely separate regions within the embedding space. This modality gap exhibits a correlation with model performance, and increasing it can somehow enhance performance in multimodal learning. To gain a deeper understanding of the performance improvements attributed to MLA, we visualize the modality gap between the text and vision modalities in the Food101 dataset in Figure 3. By comparing the performance with concatenation, MLA results in a larger modality gap, signifying that different modalities become more distinguishable and further lead to stronger performance. This further underscores the effectiveness of our proposed approach.

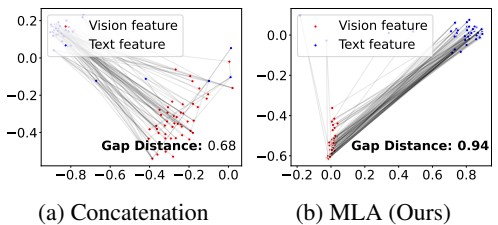

| (a) Concatenation | (b) MLA (Ours) |

Figure 3: Visualizations of the modality gap distance on the Food101 dataset. Here, both MLA and concatenation methods are depicted.

### 3.3.4 ROBUSTNESS ANALYSIS OF THE PRE-TRAINED MODEL

We evaluate the robustness of pre-trained models in this section. To do so, we utilize pre-trained features using Contrastive Language-Image Pretraining (CLIP) (Radford et al., 2021). Specifically, we employ CLIP pre-trained encoders in the Food-101 dataset and apply MLA on this newly pre-trained encoders and present the results in Table 4. From Table 4, it is evident that MLA effectively mitigates modality laziness. This is demonstrated by a significant improvement in the performance of the subordinate visual modality, while simultaneously enhancing the fusion performance. These findings underscore the effectiveness of our method in multimodal representation learning.

Table 4: Results on the Food-101 dataset achieved by changing the encoders to the CLIP pre-trained model. We report the average test accuracy (%) from three different seeds (refer to Appendix A.4 for full results).

| Method | Food-101 | | |
|---|---|---|---|
| | Image | Text | Multi |
| CLIP | 63.07 | 83.98 | 93.07 |
| **CLIP + MLA (Ours)** | **72.22** | **85.34** | **93.47** |

## 4 RELATED WORK

**Imbalanced Modality Contributions to Multimodal Learning.** Leveraging multiple modalities that contain unique yet complementary representations is crucial for understanding and solving the real world problems (Liang et al., 2022; Alwassel et al., 2020; Chen et al., 2022). However, training on multimodal data simultaneously is challenging since different modalities have significant discrepancies in their data distributions, learning architectures, and target tasks. These varying characteristics in multiple modalities hinder the model from integrating the knowledge from different senses and also lead to the problem of *modality laziness* (Du et al., 2023b; Wang et al., 2020a; Winterbottom et al., 2020; Sun et al., 2021), a phenomenon where some modalities appear more dominant than others during multimodal learning. To tackle this critical problem and utilize rich information from subordinate modalities, several approaches have recently been suggested (Wang et al., 2020b; Ismail et al., 2020b; Huang et al., 2022; Fan et al., 2023). Representatively, OGM-GE (Peng et al., 2022) proposes a balanced multimodal learning method that corrects the contribution imbalance of different modalities by encouraging intensive gradient updating from suboptimal modalities. QMF (Zhang et al., 2023) introduces a robust multimodal learning method that mitigates the impact of low-quality or noisy modalities by estimating the energy-based score of each modality.

While these methods mitigated imbalanced modality contributions to multimodal learning, they require simultaneous optimization of multiple modules for different sensory data by design, which can result in undesired interference in training on different modalities. Instead, our proposed method can avoid this inherent problem by rethinking multimodal learning to alternating unimodal learning, where the model can independently learn the corresponding unimodal information at each training step and capture the abundant cross-modal information. This paradigm allows the model to optimize each modality sufficiently, leading to building a balanced and effective multimodal model.

**Learning with Missing Modalities.** Some modality data is often missing in real-world scenarios for various reasons, such as cost, accidents, or privacy concerns. Such missing modality issue (Hotelling, 1992; Zhang et al., 2021; Ma et al., 2021; 2022; Zhang et al., 2022c) is also a crucial challenge for balanced training on multimodal data, which can be regarded as an extreme case of modality laziness. To resolve this problem, one conventional approach is a data imputation strategy (Zhao et al., 2021; Tran et al., 2017), which generates estimated data/embedding based on statistics from existing multimodal data pairs. This approach often works well on small data, but is often inappropriate for large data or when large chunks of data are missing. Alternatively, several works have addressed the problem of missing modality without the need for imputation (Zhang et al., 2022a; Ma et al., 2021; 2022). Zhang et al. (2022a) introduces a new framework for partial multi-view learning, where each incoming data is associated with only one of the multiple views (i.e., modalities). They adopt a generative adversarial strategy to estimate missing data and improve the quality of multi-view representation. SMIL (Ma et al., 2021) addresses the problem of multimodal learning with severely missing modalities (e.g., 90%). It utilizes a Bayesian meta-learning framework that learns to perturb the latent feature space and then estimates the representation of missing modalities from accessible sensory data. However, such previous works are tailored to mitigate the problem of missing modality in multimodal learning, resulting in their limited generality on diverse multimodal problems. Unlike previous works, our proposed MLA is broadly applicable to traditional multimodal learning tasks as well as challenging scenarios with missing modality data, and we have demonstrated its effectiveness by outperforming solid baselines on these tasks.

## 5 CONCLUSION

In this paper, we propose MLA to address the challenge of imbalance optimization across different modalities in multimodal learning, which is referred to as "modality laziness." Specifically, our main idea is decomposing the traditional joint optimization strategy in multimodal learning into an alternating unimodal optimization strategy. To capture cross-modal information in the unimodal optimization process, we propose a gradient modification mechanism to avoid modality forgetting. Furthermore, during test time, MLA incorporates a dynamic fusion mechanism to enable effective multimodal fusion. MLA is also compatible with scenarios including learning with complete or missing modalities. The empirical results demonstrate the promise of MLA in mitigating modality laziness and further improving performance under five datasets with different types of modalities.

## REPRODUCIBILITY STATEMENT

In the context of our empirical findings, we elaborate on the specifics of the datasets employed in our study and provide a comprehensive enumeration of hyperparameters for both the complete multimodal learning scenario and the scenario involving missing modalities. These details can be found in Appendix A.3.1 and A.3.2, respectively.We will open-source the code upon publication.

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

## A  APPENDIX

### A.1  DETAILED DATASET DESCRIPTION

**CREMA-D** (Creole Multimodal Affect Database): CREMA-D is a multimodal dataset designed for emotion recognition research. It contains audio and video recordings of actors from the Haitian Creole culture portraying various emotional states. The dataset is valuable for studying the cross-cultural aspects of emotion recognition and includes a range of emotions expressed through speech and facial expressions.

**Kinetic-Sound**: Kinetic-Sound is a dataset that combines the fields of computer vision and audio processing. It consists of synchronized video and audio recordings of everyday objects and actions, providing a rich resource for research on audio-visual scene understanding and object recognition. Researchers use Kinetic-Sound to explore how audio and visual information can complement each other for better recognition and understanding of real-world scenes.

**Food-101**: Food-101 is a widely-used dataset for food image classification and recognition. It contains over 100,000 images of 101 different food categories, making it a valuable resource for training and evaluating machine learning models for food recognition tasks. Researchers and developers use Food-101 to build applications for automated food identification, dietary analysis, and more.

**MVSA** (Multimodal Visual Sentiment Analysis): MVSA is a dataset designed to study sentiment analysis in multimedia content. It combines text, image, and audio data to capture the sentiment expressed in social media posts. Researchers use MVSA to develop advanced models for understanding the emotions and sentiments conveyed in multimedia content, which is crucial for applications like social media monitoring and sentiment analysis in marketing.

**IEMOCAP** (Interactive Emotional Dyadic Motion Capture): IEMOCAP is a unique dataset created for research in emotion recognition and speech processing. It consists of audio and motion-capture data recorded during natural, emotionally charged conversations between actors. IEMOCAP is widely used to advance the understanding of emotion in speech and gesture, as well as for developing emotion-aware conversational AI systems and therapy applications.

These datasets play crucial roles in advancing research in their respective fields, providing valuable resources for developing and evaluating models and algorithms related to emotion recognition, audio-visual scene understanding, food recognition, sentiment analysis, and emotion in speech and gesture.

### A.2  DETAILS OF BASELINES

In this section, we provide a comprehensive overview of the baseline methods employed in our multimodal learning framework.

**Summation**: Summation fusion, also known as element-wise addition, serves as a fundamental technique for integrating information from diverse modalities. It combines representations from different modalities through a straightforward element-wise summation process. While this method effectively merges information from multiple sources, it does so without explicitly modeling interactions between modalities. In the summation method, each modality-specific encoder is equipped with a fully connected layer. The input and output dimensions of this layer correspond to the output dimension of the encoder and the number of classes, respectively. Subsequently, the unimodal outputs from these fully connected layers are summed to obtain a fusion output. The fusion output is then utilized to calculate the loss and update all parameters of the modality-specific encoders and fully connected layers.

**Concatenation**: Concatenation fusion involves the concatenation of feature vectors from different modalities along a specific axis. This technique enables the model to perceive combined information but does not inherently capture cross-modal interactions. It serves as a foundational approach for feeding multimodal data into neural networks. In the concatenation method, a single fully connected layer is employed. The input dimension of this layer equals the sum of the output dimensions of all encoders, while the output dimension corresponds to the number of classes. During forward propagation, all encoder outputs are concatenated and passed into the fully connected layer to obtain

the fusion output. During backpropagation, the loss computed using the fusion output and ground truth is used to update all parameters, including those of the encoders and the fully connected layer.

**Late fusion (Gunes & Piccardi, 2005):** Late fusion approaches involve processing each modality separately and subsequently combining the results at a later stage. This method provides flexibility in handling modalities independently before fusing them, but may miss capturing early interactions between data sources. In the late fusion method, multiple fully connected layers correspond to each modality-specific encoder, similar to the summation approach. However, in this method, there are no cross-modal interactions. All encoders and their corresponding fully connected layers are trained independently on a single modality. The fusion output is calculated as the average of all unimodal outputs.

**FiLM (Perez et al., 2018):** FiLM is an advanced multimodal fusion technique that introduces condition encoding and feature modulation. By modulating visual features based on textual conditions, FiLM enables dynamic adjustments of visual representations in response to textual inputs. This method is particularly effective in tasks requiring fine-grained alignment between modalities. In the FiLM method, either audio or text serves as the condition encoder's input to conditionally encode the remaining modalities. The condition encoder is implemented as a fully connected layer, with its input dimension matching the output dimension of the modality-specific encoder. The output dimension of the condition encoder is twice its input dimension, and the encoding is split into $\gamma$ and $\beta$ components, which are used to modify the output of other modalities. The modified output is then appended to a fully connected layer for classification and loss computation during optimization.

**BiGated (Kiela et al., 2018):** BiLinear Gated Fusion leverages bilinear pooling and gating mechanisms to capture intricate interactions between modalities, explicitly modeling cross-modal correlations. This approach provides a more expressive fusion strategy for capturing nuanced relationships between different data sources. In the BiGated method, multiple fully connected layers correspond to each encoder, similar to the summation approach. Additionally, a fusion layer aggregates the outputs from all modalities, as in the concatenation method. However, one modality's hidden state undergoes an activation function (sigmoid in our experiments) to derive a gated weight. This weight is used to modulate the other hidden states before they are appended to the fully connected layer to obtain the fusion output.

**OGM-GE (Peng et al., 2022):** OGM-GE addresses under-optimization for specific modalities in multimodal learning by modifying the gradients' values. This approach balances attention weights for each modality, reducing the impact of less informative modalities on the fusion output during training. In the OGM-GE method, which is implied in the setting of concatenation, modality-specific coefficients $\kappa_t^u$ are calculated according to the algorithm outlined in the original paper. These coefficients are then used to scale all gradients during backpropagation, balancing the optimization of each modality.

**QMF (Zhang et al., 2023):** Quality-aware Multimodal Fusion (QMF) is designed to efficiently fuse various modalities, particularly in information imbalance situations. Similar to late fusion, QMF introduces a multimodal loss to learn cross-modal information and a regularization term to encourage modalities to pay more attention to challenging samples. In our experiments, we applied QMF similarly to the late fusion setting. The loss calculation aligns with the methodology described in the QMF papers.

## A.3 DETAILS OF EXPERIMENTAL SETUP

### A.3.1 LEARNING WITH COMPLETE MODALITY

To evaluate the effectiveness of our method across diverse models and datasets, we employed three distinct encoders:

**ResNet-18 Based Network:** We utilized a ResNet-18 based network for the CREMA-D and KS datasets. The ResNet-18 architecture is a member of the ResNet family, specifically designed to address the challenge of vanishing gradients during the training of deep neural networks. It introduces the concept of residual connections, also known as skip connections or shortcut connections, which facilitate the direct flow of information across network layers. This alleviates the vanishing gradient issue, enabling the training of exceptionally deep networks. In our experiments, we initialized the weights of ResNet-18 using the standard initialization method.

**M3AE (Multimodal Multimodal Contrastive Learning Based Encoder):** For the Food-101 and MVSA datasets, we employed the M3AE encoder. M3AE is a large-pretrained model designed for both vision and language data, leveraging multimodal contrastive learning. It has demonstrated outstanding performance in various downstream tasks. We initiated the M3AE encoder by loading its base model, which has been released publicly.

**M3AE + CAVMAE:** In the case of the IEMOCAP dataset, we adopted a combination of encoders. The acoustic encoder was based on CAVMAE (Audio-Vision Task Pretrained Encoder), a large-pretrained model specialized for audio-vision tasks. For the visual-textual modality, we utilized M3AE. The weight initialization for the CAVMAE encoder was performed by loading the pre-trained cavmae-audio model.

In all experiments, we employed a shared head consisting of a fully connected layer. The input dimension for this layer was set to 512 for experiments with the base model and 768 for experiments with the large-pretrained model. The output dimension of this fully connected layer matched the number of classes specific to each dataset. We employed the Stochastic Gradient Descent (SGD) optimizer for all experiments with a momentum value of 0.9. The initial learning rate was set to 0.001. Learning rate decay was applied at regular intervals during training, with a decay ratio of 0.1. In all experiments, we utilized a batch size of 64.

The choice of features varied across datasets and modalities: For the CREMA-D and KS datasets, we utilized the fbank acoustic feature, while the visual feature consisted of a concatenation of three transformed images. In the case of the Food-101, MVSA, and IEMOCAP datasets: Textual Feature: We employed tokenized text extracted using a BERT-based model. Visual Feature: For these datasets, the visual feature was based on transformed images. Acoustic Feature (IEMOCAP only): For the IEMOCAP dataset, the acoustic feature was incorporated into the CAVMAE encoder and consisted of the fbank feature.

### A.3.2 LEARNING WITH MISSING MODALITIES

In our experiments focusing on missing modalities within the IEMOCAP dataset, we employed a diverse set of feature extraction models to capture textual, visual, and acoustic information. Here, we provide detailed descriptions of the feature extraction models and their respective advantages:

We extracted textual features using BERT (Bidirectional Encoder Representations from Transformers), a groundbreaking natural language processing model developed by Google AI in 2018. BERT revolutionized the field of NLP by introducing a pre-trained model capable of understanding contextual relationships in both directions (left-to-right and right-to-left) within a sentence. This bidirectional approach enables BERT to capture intricate word relationships, making it highly effective for a wide range of NLP tasks, including text classification, question-answering, and sentiment analysis.

For visual feature extraction, we relied on MANet, an advanced pre-trained model specializing in visual tasks. MANet builds upon the success of convolutional neural networks (CNNs) by incorporating multi-attention mechanisms. This unique design allows MANet to efficiently process and understand visual information by selectively attending to relevant regions within an image. Consequently, MANet excels in tasks such as image recognition, object detection, and scene understanding, enabling it to capture detailed context and intricate visual relationships.

To extract acoustic features, we utilized Wav2vec, an innovative pre-trained model developed by Facebook AI in 2019. Wav2vec is specifically designed for speech and audio processing tasks, offering the capability to directly convert raw audio waveforms into meaningful vector representations. This model has significantly enhanced the accuracy and efficiency of various audio-related applications, including automatic speech recognition, voice activity detection, and audio classification.

For all methods, we employed modality-specific encoders consisting of a sequence of three fully connected layers. The input dimensions for the acoustic, visual, and textual modalities were set to 512, 1024, and 1024, respectively. The embedding size for all fully connected layers was fixed at 128. The classifier utilized in all methods was a fully connected layer. Both the input and output dimensions of the classifier were set to 128 and corresponded to the number of classes specific to the dataset. The batch size for all experiments was set to 64. We initiated training with an initial learning rate of 0.001, which was reduced in every iteration with a decay ratio of 0.1.

This comprehensive setup allowed us to explore the impact of missing modalities in the IEMOCAP dataset while leveraging state-of-the-art feature extraction models for textual, visual, and acoustic data. The combination of BERT, MANet, and Wav2vec, along with carefully tuned network architectures and training parameters, enabled us to conduct rigorous and insightful experiments in multimodal learning.

## A.4 FULL RESULTS

### A.4.1 FULL RESULT OF COMPLETE MULTIMODAL LEARNING

Table 5: The full results on audio-video (A-V), image-text (I-T), and audio-image-text (A-I-T) datasets. Both the results of only using a single modality and the results of combining all modalities ("Multi") are listed. We report the average test accuracy (%) of three random seeds. The best results and second best results are **bold** and underlined, respectively.

| Type | Data | | Sum | Concat | Late Fusion | FiLM | BiGated | OGM-GE | QMF | MLA (Ours) |
|---|---|---|---|---|---|---|---|---|---|---|
| A-V | CREMA-D | Audio | $54.14 \pm 0.92$ | $55.65 \pm 0.82$ | $52.17 \pm 1.12$ | $53.89 \pm 0.75$ | $51.49 \pm 1.28$ | $53.76 \pm 0.98$ | $\mathbf{59.41 \pm 0.71}$ | $\underline{59.27 \pm 1.23}$ |
| | | Video | $18.45 \pm 1.07$ | $18.68 \pm 0.76$ | $\underline{55.48 \pm 0.71}$ | $18.67 \pm 1.21$ | $17.34 \pm 1.11$ | $28.09 \pm 1.17$ | $39.11 \pm 1.03$ | $\mathbf{64.91 \pm 1.10}$ |
| | | Multi | $60.32 \pm 0.78$ | $61.56 \pm 0.91$ | $66.32 \pm 1.08$ | $60.07 \pm 1.25$ | $59.21 \pm 1.14$ | $\underline{68.14 \pm 0.79}$ | $63.71 \pm 1.12$ | $\mathbf{79.70 \pm 0.87}$ |
| | KS | Audio | $48.77 \pm 0.95$ | $49.18 \pm 0.76$ | $47.87 \pm 1.10$ | $48.67 \pm 0.83$ | $49.96 \pm 1.21$ | $48.87 \pm 1.05$ | $\underline{51.57 \pm 1.03}$ | $\mathbf{54.67 \pm 0.92}$ |
| | | Video | $24.53 \pm 1.12$ | $24.67 \pm 1.07$ | $\underline{46.76 \pm 0.85}$ | $23.15 \pm 0.98$ | $23.77 \pm 1.14$ | $29.73 \pm 1.06$ | $32.19 \pm 0.92$ | $\mathbf{51.03 \pm 1.09}$ |
| | | Multi | $64.72 \pm 0.97$ | $64.84 \pm 1.05$ | $65.53 \pm 0.89$ | $63.33 \pm 0.76$ | $63.72 \pm 1.13$ | $65.74 \pm 1.08$ | $\underline{65.78 \pm 0.97}$ | $\mathbf{71.35 \pm 1.22}$ |
| I-T | Food-101 | Image | $4.57 \pm 0.88$ | $3.51 \pm 1.22$ | $\underline{58.46 \pm 1.03}$ | $4.68 \pm 0.97$ | $14.20 \pm 1.18$ | $22.35 \pm 1.27$ | $45.74 \pm 1.09$ | $\mathbf{69.60 \pm 0.89}$ |
| | | Text | $85.63 \pm 1.14$ | $\underline{86.02 \pm 1.05}$ | $85.19 \pm 1.21$ | $85.84 \pm 0.92$ | $85.79 \pm 1.10$ | $85.17 \pm 1.09$ | $85.07 \pm 1.03$ | $\mathbf{86.47 \pm 0.86}$ |
| | | Multi | $86.19 \pm 0.86$ | $86.32 \pm 1.18$ | $90.21 \pm 1.02$ | $87.21 \pm 1.05$ | $88.87 \pm 0.88$ | $87.54 \pm 1.03$ | $\underline{92.87 \pm 1.01}$ | $\mathbf{93.33 \pm 0.92}$ |
| | MVSA | Text | $73.33 \pm 1.03$ | $\underline{75.22 \pm 0.92}$ | $72.15 \pm 1.07$ | $74.85 \pm 0.98$ | $73.13 \pm 1.08$ | $74.76 \pm 0.87$ | $74.87 \pm 1.15$ | $\mathbf{75.72 \pm 0.79}$ |
| | | Image | $28.46 \pm 1.12$ | $27.32 \pm 1.09$ | $\underline{45.24 \pm 0.97}$ | $27.12 \pm 1.18$ | $28.15 \pm 0.88$ | $31.98 \pm 1.03$ | $32.99 \pm 1.09$ | $\mathbf{54.99 \pm 1.12}$ |
| | | Multi | $76.19 \pm 1.01$ | $76.25 \pm 0.95$ | $76.88 \pm 0.82$ | $75.34 \pm 1.07$ | $75.94 \pm 0.88$ | $76.37 \pm 0.98$ | $\underline{77.96 \pm 1.06}$ | $\mathbf{79.94 \pm 0.97}$ |
| A-I-T | IEMOCAP | Audio | $39.79 \pm 1.08$ | $41.93 \pm 0.89$ | $\underline{43.12 \pm 0.97}$ | $41.64 \pm 1.06$ | $42.23 \pm 0.88$ | $41.38 \pm 1.13$ | $42.98 \pm 0.95$ | $\mathbf{46.29 \pm 1.02}$ |
| | | Image | $29.44 \pm 0.97$ | $30.00 \pm 0.88$ | $\underline{32.38 \pm 0.92}$ | $29.85 \pm 1.06$ | $27.45 \pm 1.08$ | $30.24 \pm 1.02$ | $31.22 \pm 1.07$ | $\mathbf{37.63 \pm 0.78}$ |
| | | Text | $65.16 \pm 0.88$ | $67.84 \pm 0.97$ | $68.79 \pm 1.07$ | $66.37 \pm 0.95$ | $65.16 \pm 1.08$ | $\underline{70.79 \pm 1.03}$ | $75.03 \pm 0.79$ | $\mathbf{73.22 \pm 1.09}$ |
| | | Multi | $74.18 \pm 1.03$ | $75.91 \pm 0.92$ | $74.96 \pm 0.97$ | $74.32 \pm 1.12$ | $73.34 \pm 1.05$ | $\underline{76.17 \pm 1.01}$ | $\underline{76.17 \pm 0.95}$ | $\mathbf{78.92 \pm 1.07}$ |

### A.4.2 FULL RESULTS OF ABLATION STUDIES

In Table 6, we report the full results of ablation studies with 95% standard deviation.

Table 6: We report the test accuracy percentages (%) on the IEMOCAP dataset using three different seeds, while applying varying modality missing rates to audio, image, and text data. The best results are highlighted in **bold**, while the second-best results are underlined.

| Method | Modality Missing Rate (%) | | | | | | |
|---|---|---|---|---|---|---|---|
| | 10 | 20 | 30 | 40 | 50 | 60 | 70 |
| Late Fusion | $72.95 \pm 1.12$ | $69.06 \pm 0.88$ | $64.89 \pm 1.25$ | $61.09 \pm 0.99$ | $56.48 \pm 1.18$ | $52.41 \pm 1.01$ | $45.07 \pm 1.21$ |
| QMF | $\underline{73.49 \pm 1.10}$ | $\underline{71.33 \pm 1.27}$ | $65.89 \pm 0.78$ | $62.27 \pm 0.95$ | $57.94 \pm 1.07$ | $55.60 \pm 0.84$ | $50.25 \pm 1.09$ |
| CCA | $65.19 \pm 0.95$ | $62.60 \pm 1.19$ | $59.35 \pm 0.88$ | $55.25 \pm 1.23$ | $51.38 \pm 1.05$ | $45.73 \pm 1.27$ | $30.61 \pm 1.10$ |
| DCCA | $57.25 \pm 1.28$ | $51.74 \pm 1.01$ | $42.53 \pm 1.29$ | $36.54 \pm 1.18$ | $34.82 \pm 0.79$ | $33.65 \pm 1.07$ | $41.09 \pm 1.24$ |
| DCCAE | $61.66 \pm 1.13$ | $57.67 \pm 1.22$ | $54.95 \pm 1.06$ | $51.08 \pm 1.01$ | $45.71 \pm 0.87$ | $39.07 \pm 0.98$ | $41.42 \pm 1.28$ |
| AE | $71.36 \pm 0.94$ | $67.40 \pm 0.79$ | $62.02 \pm 1.18$ | $57.24 \pm 0.94$ | $50.56 \pm 0.77$ | $43.04 \pm 0.96$ | $39.86 \pm 0.82$ |
| CRA | $71.28 \pm 1.19$ | $67.34 \pm 1.04$ | $62.24 \pm 1.03$ | $57.04 \pm 0.92$ | $49.86 \pm 1.15$ | $43.22 \pm 0.99$ | $38.56 \pm 1.22$ |
| MMIN | $71.84 \pm 0.97$ | $69.36 \pm 1.05$ | $\underline{66.34 \pm 1.11}$ | $\underline{63.30 \pm 1.10}$ | $\underline{60.54 \pm 1.17}$ | $57.52 \pm 1.13$ | $\underline{55.44 \pm 1.06}$ |
| IF-MMIN | $71.32 \pm 0.78$ | $68.29 \pm 0.98$ | $64.17 \pm 1.01$ | $60.13 \pm 1.19$ | $57.45 \pm 1.12$ | $53.26 \pm 1.22$ | $52.04 \pm 0.91$ |
| CPM-Net | $55.29 \pm 0.83$ | $53.65 \pm 0.96$ | $52.52 \pm 1.29$ | $51.01 \pm 0.94$ | $49.09 \pm 1.04$ | $47.38 \pm 1.25$ | $44.76 \pm 0.88$ |
| TATE | $67.84 \pm 1.02$ | $63.22 \pm 1.08$ | $62.19 \pm 0.98$ | $60.36 \pm 1.06$ | $58.74 \pm 1.21$ | $\underline{57.99 \pm 1.03}$ | $54.35 \pm 1.07$ |
| **MLA (Ours)** | $\mathbf{75.07 \pm 1.04}$ | $\mathbf{72.33 \pm 1.16}$ | $\mathbf{68.47 \pm 1.22}$ | $\mathbf{67.00 \pm 0.98}$ | $\mathbf{63.48 \pm 0.87}$ | $\mathbf{59.17 \pm 0.92}$ | $\mathbf{55.89 \pm 1.03}$ |

### A.4.3 FULL RESULTS OF CLIP

Table 7: Results on the Food-101 dataset achieved by changing the encoders to the CLIP pre-trained model. We report the average test accuracy (%) from three different seeds.

| Method | Food-101 | | |
|---|---|---|---|
| | Image | Text | Multi |
| CLIP | $63.07 \pm 0.12$ | $83.98 \pm 0.08$ | $93.07 \pm 0.08$ |
| **CLIP + MLA (Ours)** | $\mathbf{72.22 \pm 0.10}$ | $\mathbf{85.34 \pm 0.06}$ | $\mathbf{93.47 \pm 0.04}$ |

