# OpenReview forum: "Multimodal Representation Learning by Alternating Unimodal Adaptation"
_ICLR.cc/2024/Conference — ICLR 2024 Conference Withdrawn Submission_

### Official Review · Reviewer_LsYd · 2023-10-29

**Soundness:** 3 good
**Presentation:** 2 fair
**Contribution:** 2 fair
**Rating:** 5
**Confidence:** 4

**Summary:**

This paper proposes to learn multimodal representation by alternating unimodal adaptation so that the imbalance optimization across different modalities is alleviated. Experiments show that the proposed method shows better performance over some multimodal methods and is more robust when some modalities are missing.

**Strengths:**

+ This paper is clear and easy to understand.
+ The ablation study is convincing

**Weaknesses:**

- The contribution, alternating unimodal learning, is to balance the optimization of different modalities. In my understanding, it just update the unimodal model parameters one by one to achieve the "balance". How can you ensure that the unimodal parameters do not learn at different speeds as found in [a]? And also the reference for [a] is repeated in the paper.

[a] What Makes Training Multi-modal Classification Networks Hard?

- Why not comparing the proposed method to recent multimodal fusion methods, like [b][c]?
[b] Multimodal Token Fusion for Vision Transformers
[c] Everything at once-multi-modal fusion transformer for video retrieval

- The proposed dynamic fusion is simple but in the ablation, it cannot improve the multimodal performance much.

**Questions:**

See weaknesses.

---

### Official Review · Reviewer_KkKe · 2023-10-31

**Soundness:** 3 good
**Presentation:** 2 fair
**Contribution:** 2 fair
**Rating:** 5
**Confidence:** 4

**Summary:**

This paper proposes MLA (Multimodal Learning with Alternating Unimodal Adaptation) to address the issue of modality laziness in multi-modal settings and validates the effectiveness of the method on multiple datasets.

**Strengths:**

1. This paper combines gradient modification and a test-time uncertainty-based model fusion mechanism to address the classic problem in multi-modal learning: Modality Laziness.

2. The method not only improves performance compared to the baseline but is also suitable for scenarios involving modality missing.

**Weaknesses:**

1. Table 1 is quite confusing. Why are there results for multi-modal models when only audio or video data is used? In my understanding, if only single-modality is used, there should be no results for late fusion/FiLM/OGM-GE. Are the results for audio and video derived solely from encoder evaluation using different multi-modal training methods?

2. Is a shared head required? In my understanding, each modality can have its own head, and gradient modification can be calculated in the output logits, with test-time dynamic fusion during testing.

3. The comparison baseline is limited, for example, lacking Uni Modal Ensemble[1].  Even in my understanding, if the shared head doesn't play a significant role, the method proposed in this paper can be straightforwardly replaced by a Uni-Modal Ensemble coupled with test-time dynamic fusion.

4. It appears that the output of each modality's encoder needs to have the same dimension to match the shared head's dimension, significantly reducing the overall flexibility of the learning framework.

[1] On uni-modal feature learning in supervised multi-modal learning

**Questions:**

See weakness.

---

### Official Review · Reviewer_FTB4 · 2023-11-01

**Soundness:** 2 fair
**Presentation:** 2 fair
**Contribution:** 3 good
**Rating:** 5
**Confidence:** 3

**Summary:**

In this paper, the authors propose a new multimodal learning method MLA that transforms the problem into an alternating unimodal learning process with a shared head network to avoid the modality laziness issue. To prevent the shared head from forgetting previously learned modality information, a gradient modification method is proposed to calibrate the weight update direction of the shared head. During inference, the final prediction is calculated as the weighted sum of all unimodal predictions, where the weights are calculated using the uncertainty (entropy) of each unimodal prediction. Experiments show that the proposed method outperforms other baselines in multiple multi-modal datasets and missing modality scenarios.

**Strengths:**

1. Solving the modality laziness issue is an important problem in multimodal learning
2. The proposed alternating unimodal learning is interesting and novel. It could successfully reduce modality laziness and is naturally resistant to missing modality scenarios

**Weaknesses:**

1. I am concerned regarding the training process of the proposed method. Training different modalities alternatively may make it harder for the loss function to converge. It would be better to include sample training curves to clarify this point. Besides, according to equation (3), the update of the modification matrix P_t requires calculating the average feature h_{m_t} over the entire dataset, which may greatly increase the total training time.

2. The experiment section is not well-written and causes several confusions (see questions)

**Questions:**

1. In Table 1, could the authors provide more details about how to get the uni-modal results for non-late-fusion methods such as sum and concat? Does it involve training another head network for the uni-modal encoder or simply disabling one modality in the multi-modal network?

2. In Table 3, the results without HGM and DF are exactly the same as the Late Fusion results in Table 1. Is that a mistake or a typo? For the proposed method, even without HGM and DF, it should still be different from Late Fusion because it has a shared head for all modalities and it is trained alternatively among different modalities. Could the authors provide more explanations on this?

3. In section 3.3.3, the authors show that the proposed method provides larger modality gaps and hence leads to stronger performance. However, concatenation seems to be a weak baseline. Does MLA have larger modality gaps than recent methods such as QMF? Also, from [1], increasing the modality gap may not necessarily lead to better performance. Sometimes shrinking the modality gap can also improve the performance. Could the authors provide more discussions on that?


[1] Liang, Victor Weixin, et al. "Mind the gap: Understanding the modality gap in multi-modal contrastive representation learning." Advances in Neural Information Processing Systems 35 (2022): 17612-17625.

---

### Official Review · Reviewer_BKPc · 2023-11-01

**Soundness:** 3 good
**Presentation:** 3 good
**Contribution:** 2 fair
**Rating:** 5
**Confidence:** 4

**Summary:**

This paper proposes a method for multimodal representation learning based on alternating algorithms. The writing is good. The experiment is sufficient. However, some SOTA methods are not cmpared.

**Strengths:**

The writing is good. The experiment is sufficient.

**Weaknesses:**

Some SOTA methods are not cmpared, such as SMIL (Ma et al., 2021).

**Questions:**

In this paper, the authors mention the problem of missing modalities and also conduct experiments. However, the method of SMIL (Ma et al., 2021) is not compared.

---

### Official Review · Reviewer_7Vty · 2023-11-03

**Soundness:** 3 good
**Presentation:** 2 fair
**Contribution:** 2 fair
**Rating:** 3
**Confidence:** 4

**Summary:**

This paper addresses the challenge of modality laziness, which occurs when certain modalities are not fully optimized while jointly optimizing multimodal data. The proposed approach tackles this issue by processing only one modality per iteration, employing a shared head, and utilizing distinct encoders for each modality. To mitigate the introduced modality forgetting problem, the paper introduces a gradient modification matrix. Furthermore, the method incorporates test-time modality fusion to enhance inference performance. Experimental results demonstrate that this approach surpasses jointly optimized methods in both single modality and multimodality scenarios.

**Strengths:**

(1) The utilization of alternative adaptations for each modality effectively mitigates the modality laziness issue.

(2) Good experimental results are achieved on multi-modality data to validate the effectiveness of the method.

**Weaknesses:**

(1) Further clarification is needed for the following points:
a) The initialization of the modification matrix (P_1) is not specified.
b) For the equation (3), the dividend of q_t should be P_t or P_(t-1)?
c) The setting for using a single modality is unclear. Does it pertain to training with a single modality or inferring with a single modality?

(2) In Table 4, it is not specified whether the CLIP method refers to a pre-trained CLIP or a fine-tuned CLIP.

(3) The paper states that all modalities share a head, and mentions that "the output dimension of this fully connected layer matched the number of classes specific to each dataset." Does this imply that the label space for each dataset is fixed? The question seems too narrow and the application may be limited.

(4) Regarding the novelty of the work, the paper's contributions appear to be somewhat limited, as the key techniques such as the gradient-based technique to avoid forgetting have long been studied in the literature on continual learning including orthogonal weights modification (OWM) [1], Orthogonal gradient descent (OGD) [2], and PCA-OGD [3].

(5). This paper uses a special case of the OWM method when the number of fully connected layers is 1. The ablation study of the architecture of the shared head and the selection of this gradient-modification technique is missing.

(6) The paper makes an assumption that inferior modality gives high-entropy predictions. Experiments to examine or support this assumption are missed.

Reference List
[1] Zeng, Guanxiong, et al. "Continual learning of context-dependent processing in neural networks." Nature Machine Intelligence 1.8 (2019): 364-372.
[2] Farajtabar, Mehrdad, et al. "Orthogonal gradient descent for continual learning." International Conference on Artificial Intelligence and Statistics. PMLR, 2020.
[3] Doan, Thang, et al. "A theoretical analysis of catastrophic forgetting through the ntk overlap matrix." International Conference on Artificial Intelligence and Statistics. PMLR, 2021.

**Questions:**

(1). The sequence to iterate each modality is not unique. Does and how the order to iterate affect the alternating training? For example, when there is image, video, and text data. Does the sequence of Image->Video->Text differ from the sequence of Image->Text->Video?

(2). How this iteration over each modality affects the convergence rate in the unit of training iteration/step